# Detection of Antimicrobial Resistance, Pathogenicity, and Virulence Potentials of Non-Typhoidal *Salmonella* Isolates at the Yaounde Abattoir Using Whole-Genome Sequencing Technique

**DOI:** 10.3390/pathogens11050502

**Published:** 2022-04-23

**Authors:** Chelea Matchawe, Eunice M. Machuka, Martina Kyallo, Patrice Bonny, Gerard Nkeunen, Isaac Njaci, Seraphine Nkie Esemu, Dedan Githae, John Juma, Bawe M. Nfor, Bonglaisin J. Nsawir, Marco Galeotti, Edi Piasentier, Lucy M. Ndip, Roger Pelle

**Affiliations:** 1Institute of Medical Research and Medicinal Plants Studies, Yaounde P.O. Box 6163, Cameroon; bonypatrice@yahoo.com (P.B.); njuliusfrida@yahoo.com (B.J.N.); 2Bioscience Eastern and Central Africa-International Livestock Research Institute (BecA-ILRI) Hub, Nairobi P.O. Box 30709, Kenya; e.machuka@cgiar.org (E.M.M.); isaac.njac18i@jic.ac.uk (I.N.); m.kyalo@cgiar.org (M.K.); d.githae@cgiar.org (D.G.); j.juma@cgiar.org (J.J.); 3Department of Biochemistry, University of Dschang, Dschang P.O. Box 96, Cameroon; gnkeunen@gmail.com; 4Department of Microbiology and Parasitology, University of Buea, Buea P.O. Box 63, Cameroon; esemu2003@yahoo.co.uk (S.N.E.); lndip@yahoo.com (L.M.N.); 5Department of Rangeland, Animal Nutrition and Livestock Infrastructures, Sub Department of Animal Nutrition, Ministry of Livestock, Fisheries and Animal Industries (MINEPIA), Yaounde P.O. Box 930, Cameroon; nformohamadou42@yahoo.com; 6Department of Agricultural, Food, Environment and Animal Sciences, University of Udine, 33100 Udine, Italy; marco.galeotti@uniud.it (M.G.); edi.piasentier@uniud.it (E.P.)

**Keywords:** multidrug-resistance, whole-genome sequencing, non-typhoidal *Salmonella*, pathogenicity and virulence, silent resistant genes, beef carcass, Yaounde abattoir

## Abstract

One of the crucial public health problems today is the emerging and re-emerging of multidrug-resistant (MDR) bacteria coupled with a decline in the development of new antimicrobials. Non-typhoidal *Salmonella* (NTS) is classified among the MDR pathogens of international concern. To predict their MDR potentials, 23 assembled genomes of NTS from live cattle (n = 1), beef carcass (n = 19), butchers’ hands (n = 1) and beef processing environments (n = 2) isolated from 830 wet swabs at the Yaounde abattoir between December 2014 and November 2015 were explored using whole-genome sequencing. Phenotypically, while 22% (n = 5) of *Salmonella* isolates were streptomycin-resistant, 13% (n = 3) were MDR. Genotypically, all the *Salmonella* isolates possessed high MDR potentials against several classes of antibiotics including critically important drugs (carbapenems, third-generation cephalosporin and fluoroquinolone). Moreover, >31% of NTS exhibited resistance potentials to polymyxin, considered as the last resort drug. Additionally, ≤80% of isolates harbored “silent resistant genes” as a potential reservoir of drug resistance. Our isolates showed a high degree of pathogenicity and possessed key virulence factors to establish infection even in humans. Whole-genome sequencing unveiled both broader antimicrobial resistance (AMR) profiles and inference of pathogen characteristics. This study calls for the prudent use of antibiotics and constant monitoring of AMR of NTS.

## 1. Introduction

Today’s world is experiencing an antibiotic resistance pandemic due to growing bacterial resistance to a broad range of drugs in animals and clinical settings [1,2,3]. Microbial multidrug resistance (MDR) frustrates efforts for infection control resulting in a considerable increase in morbidity and mortality worldwide [1,2].

MDR has also been reported in non-typhoidal *Salmonella* (NTS). For instance, while 16% of NTS isolates exhibited resistance to at least one essential antibiotic, as high as 2% of them were resistant to at least three different classes of antibiotics in the US [2]. In Europe, 23% of NTS isolated from calf carcasses were MDR [3]. In Europe, 23% of NTS isolated from calf carcasses were MDR [3]. In Africa, there is an emergence of an invasive non-typhoidal *Salmonella* (iNTS) lineage with increased multidrug resistance (MDR) potential playing a considerable role in outbreaks, thereby threatening the global market and tourism [4,5]. Moreover, close to 80% of all the reported cases in 2017 occurred in sub-Saharan Africa alone, affecting mainly children under five, adolescents and active young people under fifty [5]. Therefore, NTS seems to be a significant cause of loss of economic activity in Africa [6]. In Cameroon, despite the absence of official statistics on foodborne diseases in general, sporadic cases of invasive salmonellosis mistaken either for malaria or typhoid fever have been recently reported [7,8,9].

Current knowledge of the type of *Salmonella* serovars and their antibiotic resistance profile is essential to inform policy and guide treatment strategies for appropriate therapy and the development of new antimicrobials [1,2]; thus, the WHO recommendation for national surveillance of antimicrobial resistance (AMR) in *Salmonella* [1]. Given their zoonotic nature, there is a need for an integrative ‘One Health’ approach for the surveillance of AMR among humans and animal *Salmonella* isolates [10]. Healthy or asymptomatic live animals such as cattle may carry *Salmonella*, thereby representing an important risk factor for beef carcass contamination during processing at the abattoirs such as the Yaoundé abattoir [11,12].

The Yaoundé abattoir where more than 6000 animals are slaughtered every week, is one of the major slaughterhouses in Cameroon that has the capacity to supply meat to three regions (centre, west and south) of Cameroon and neighboring countries (Equatorial Guinea and Gabon). Following the WHO recommendation and given the use of antibiotics for disease prevention or animal growth promotion, it seems crucial to monitor the antimicrobial resistance profile of *Salmonella* isolates at the Yaounde abattoir using molecular techniques such as whole-genome sequencing (WGS).

Unlike traditional antimicrobial susceptibility testing, whole-genome sequencing (WGS) can give information on the presence of MDR genes [13] and pathogenicity and virulence factors in *Salmonella* organisms. This study was aimed at predicting the MDR, pathogenicity, and virulence potentials of *Salmonella* isolated at the Yaounde abattoir using WGS.

## 2. Results

### 2.1. Phenotypic Antibiotic Resistance Profile of Salmonella Isolates

Twenty-three genomes of 38 identified Salmonella isolates were successfully sequenced. Only 19 sequenced genomes were thoroughly exploitable for the required bioinformatics analyses. The genomic profile of NTS isolates and their GenBank accession numbers are outlined in Appendix A.

Isolates 8ev, 20de, 22sa, 34de, 60sa, and evjul were resistant to streptomycin, whereas between 18 and 20 isolates were highly susceptible to ampicillin, chloramphenicol, and tetracycline (Table 1 and Appendix A). Interestingly, isolates 8ev, 22sa, and 34de were MDR.

### 2.2. Genotypic Antibiotic Resistance Profile of Salmonella Isolates

The distribution of specific antibiotic resistance genes among the *Salmonella* isolates is summarized in Table 2. The streptomycin-resistance genes, *aadA*, *aadA1*, and *aadA2* were, respectively, present in 15.8%, 26.3%, and 21% of *Salmonella* isolates. Moreover, chloramphenicol-resistance was found in 26.3%, 15.8%, 10.5, 10.5% of the isolates carrying *cat*, *cat1*, *cat2*, and *cat3* genes, separately while 10.5% of isolates harbored *cmlA1*, *cml5*, and *cml6* genes, respectively. The tetracycline-resistance genes, *tetA*, *tetB*, *tetC* and *tetR* were present in 5.26% 26.3%, 26.3%, 31.6% and 84.2% of isolates, respectively. Ampicillin-resistance genes *TEM-1* and *TEM-163* were identified in 15.8% and 21% of NTS isolates, respectively. However, close to 80% of *Salmonella* isolates harbored at least one false-negative result (Table 2 and Table 3).

### 2.3. Matching AST with Genotypic Resistance Potentials of Salmonella Isolates

The true positive and true negatives cases represented perfect matching between their phenotypic and genotypic antibiotic resistance (Table 3). The sensitivity was highest (62.5%) for resistance against streptomycin and chloramphenicol, and lowest for ampicillin (16.7%), respectively. Nevertheless, despite a general low sensitivity (averagely 45.3%), the specificity was 100% for all the antibiotics with the sum of sensitivity + specificity being ≥1.5 for all the tested *Salmonella* in the present study. The positive predictive value was 100% for all the tested antibiotics while the negative predictive values varied between 6.2% (for ampicillin) and 81.2% (for ampicillin).

Generally, results indicate that whole-genome sequencing predicted four times the antimicrobial resistance for NST than the traditional susceptibility testing (OR = 4.0, *p* = 0.46).

### 2.4. Multidrug Resistance Potentials of Salmonella Isolates

Eighteen genes (in purple) involved in the efflux, transport, and reduced permeability of antimicrobials were identified in *Salmonella* isolates (Table 4). Except for phenicol, the gene *TolC* was reported only in the phenotypic MDR isolates 8ev, 22sa, and 34de. The gene *golS* was detected in all the isolates. Interestingly, the MdsABC (multidrug transporter) complex was also present alongside gene *golS* in isolates that exhibited resistance potential against phenicols. The *E. coli soxS* and *soxR* genes were detected in 78.94 to 100% of *Salmonella* isolates.

Furthermore, the gene *mdtk* that promotes resistance solely against fluoroquinolone was present in 100% *Salmonella* isolates. The *AcraB* regulator gene *sdiA* was detected in all the isolates. Furthermore, sulfonamide-resistance genes *sul1* and *sul2* and the gene *CTX-M-14* that regulates resistance against third-generation cephalosporin were present in 15.8 and 5.26% of *Salmonella* isolates, respectively. Then, isolates 8ev, 22sa and 34de harbored cephalosporin-resistance genes *OXA-1*, *OXA-2*, and *OXA-7*. Additionally, the fluoroquinolone-resistance gene *qnrB1* was reported in isolates 8ev, 20de, 34de, and 60sa. The gene *macA* that mediates efflux of macrolide and secretion of enterotoxin ST11 was detected in 52.63% of NTS isolates. Moreover, 47.36% of *Salmonella* isolates hosted the gene *marA*, which exports antibiotics and disinfectants out of bacteria.

### 2.5. Polymorphism on PmrAB System Inducing Polymyxin-Resistance Potential in Salmonella Isolates

Mutations of the PmrAB system were detected in more than 31% of isolates (Table 5).

### 2.6. Pathogenicity and Virulence Factors

Our isolates had a mean probability of 94% to cause diseases in humans, though no significant difference (*p* > 0.05) was found among different *Salmonella* serovars in pathogenicity (Table 6). Moreover, their proteomes matched with a broad range of pathogenic bacterial families (466–787). Serovars Enteritidis and Poona were predicted greatest (*p* = 0.95) and least (*p* = 0.93) human pathogens, respectively.

Of the 11 identified SPIs, only C63PI was present in all the isolates (Table 7). The function of each virulence factor and each effector protein is summarized in Appendix A, respectively. Other determinants including effector proteins, adhesion factors, virulence plasmids, and toxins were detailed in Appendix A. About 82% of *Salmonella* isolates possessed SipABCD, SopBDE, SopE2, EnvEF, InvAE, Sii E, IagB, SptP, OrgB, PhoP, MisL, ShdA, and rtn, as well as cell entry-facilitating factors; between 56% and 61% of expressed adhesion factors including FimA, FimC, FimZ, HilD, ecpD1, CsgD, FliZ, FliT; also, between 56% and 69%, NTS expressed SopABDE, SpiC, SpvB, SseBCD, PipB2, SptP, SopE2, Spa family as response regulators. Additionally, while 95% of NTS carried H-NS, between 21% and 47% expressed heat shock and stress proteins, specifically CuSA, ScsC, and bacteriocin exporter. Four types of plasmids (IncF, IncI, Col (PHAD28), and IncH) and four bacterial toxins (Flavodoxin, Shiga-like toxin A, entericidin A, and Thioredoxin 1) were detected in some *Salmonella* isolates.

## 3. Discussion

Despite a relatively moderate resistance shown to streptomycin (21.7%), the majority of *Salmonella* strains isolated at the Yaounde abattoir were sensitive to tetracycline, chloramphenicol, and ampicillin (Table 1). These results corroborate the findings of previous studies that showed common resistance against streptomycin among *Salmonella* [14].

Findings from the present study indicate that tetracycline, chloramphenicol, and ampicillin are still effective antibiotics in Cameroon contrary to the situation in many countries where these drugs are no longer appropriate for the treatment of invasive salmonellosis. Notwithstanding, the presence of 13% of MDR *Salmonella* isolates constitutes a serious health concern. Similarly, MDR was recently reported to be observed among *Salmonella* isolated in an Ethiopian abattoir on similar drugs [15]. In the present work, MDR salmonellae were isolated from beef at different processing steps underlining a significant food safety hazard. The development and spread of AMR among NTS are specifically crucial when found in environmental settings such as the abattoir. Such an environment may be a source of cross-contamination between meat products and meat handlers to become a threat to public health.

Furthermore, the detection of the respective resistance genes to streptomycin, chloramphenicol, tetracycline, and ampicillin in the *Salmonella* isolates confirms their phenotypic antimicrobial resistance status described in Table 1. However, false-negative cases or “silent genes” were reported for each of the tested antibiotics. They represent genes that were previously detected in the susceptible isolates but failed to be expressed phenotypically. Previous studies attributed this mismatch between the phenotypic and genotypic resistance profile of bacteria to the fact that such genes were in “silent mode” in vitro [16,17,18,19]. The resistance phenotype depends on the mode and level of expression of the resistance gene that could be influenced by growth or environmental factors [16]. However, the reasons for their no phenotypic expression are not yet fully elucidated based on previous studies [16,17,18,19]. The absence or impairment of the promoter sequence or the presence of negative regulators might have downregulated their expression in vitro. Alternatively, the expression of these genes might have occurred without the expression of the gene products. Therefore, many factors exist to control the expression of the resistance phenotypes. However, if by any mechanism these genes are switched onto “activated mode”, the host bacteria will modify their phenotypic resistance status [20]. Consequently, “silent genes” can be regarded as a reservoir of antimicrobial resistance for major foodborne bacteria via horizontal gene transfer or other mechanisms [17]. This hypothesis supports previous works that considered “silent genes” as a significant potential threat to the therapeutic efficacy of antibiotics [17,19].

Furthermore, the non-expression of the “silent genes” in the susceptible phenotypes underscores a lack of definite genotype–phenotype correlation testing indicating a lack of clear genotype–phenotype correlation (OR = 4.0; *p* = 0.46). However, the value of greater than 1.5 of the sum of sensitivity and specificity reflects the usefulness of the WGS prediction (Table 3). The high PPV predicts a high prevalence of antimicrobial resistance among NTS isolates at the Yaounde abattoir. Similarly, the absence of a conclusive correlation between the phenotypic and genotypic characteristics was also reported among non-typhoidal *Salmonella* isolated from wildlife [13]. These findings clearly show that antibiotic susceptibility testing (AST) does not provide any information about the underlying genes responsible for the resistance. This suggests that WGS could be used as a complementary tool to AST to extract additional information such as the presence of silent resistance genes.

In addition, WGS is also useful in predicting the multiple drug resistance of NTS. Based on their mode of action, the eighteen MDR-promoting genes detected in this study are grouped into: (i) those that export drugs out of the bacterial cells (*sdiA*, *mdsA*, *mdsB*, *mdsC*, *E. coli soxS*, *E. coli soxR*, *acrB*, *acrD*, *golS*, *marA*, *patA* and *ramR*); (ii) those that reduce membrane permeability to drugs (*E. coli soxR*, *E. coli soxS*, and *marA*); and (iii) those that alter antibiotic target configuration (*bacA*, *E. coli soxR*, *E. coli soxS*, and *ramA*) [17]. Nevertheless, the mechanism of action of some genes, such as *E. coli sox R*, *E. coli soxS*, and *ramA* overlap (Data not shown).

All the genes involved in drug efflux are part of the resistance nodulation cell division (RND) efflux systems and effectively perform their duty in synergy with *TolC* [19]. Moreover, there were also multi-efflux pumps such as *mdsA*, *mdsB*, *mdsC*, *acrA*, acrB, and *mdtk*, which generally work in synergy either with transcriptional activators (*E. coli soxS* and *E. coli soxR*, *ramA*, *marA*) or promotors such as *TolC* and *golS* [20]. The presence of *TolC* only in MDR isolates justifies the critical role it plays in synergy with RND efflux systems in exporting a range of antimicrobials. The *golS* gene promotes the MdsABC complex to disseminate resistance against a variety of drugs and toxins and confers virulence and pathogenicity potentials to *Salmonella* [21]; thus, the presence of the MdsABC complex in all the isolates that had *golS*. The gene *sdiA*, a regulator for *AcraB*, a multi-drug resistance pump [22] detected in 100% of the isolates is a powerful promoter of resistance against a vast arsenal of antibiotics (data not shown).

The presence of *mdtK* and *qnrB1* in our isolates is extremely crucial because they could synergistically offer *Salmonella* an advantage to develop resistance via a plasmid-mediated or efflux pump mechanism against fluoroquinolone [13,23]. However, transcriptional activators, particularly *ramR*, *soxS* and *marA* could equally induce resistance against fluoroquinolone via overexpression of *acrAB*-TolC efflux pump in *Salmonella* [24]. The detection of *OXA-1*, *OXA-2*, *OXA-7*, *CTX-M-14*, and *qnrB* in some isolates is critical because they, respectively, confer resistance against third-generation cephalosporin and fluoroquinolone, all considered as the WHOs highest priority drugs [25,26]. Particularly, *OXA-1* and *OXA-2* genes exhibit broad-spectrum cephalosporin-hydrolyzing and carbapenem-hydrolyzing activities, respectively [27,28]. Furthermore, the resistance potential to polymyxin is crucial because of its consideration as the last resort drug.

Polymyxin is a bactericidal polypeptide, which disrupts lipid A subunit of the LPS outer membrane of Gram-negative bacteria causing cell lysis and their eventual irreversible death. One of the key resistance mechanisms to polymyxin adapted by *Salmonella* resides in the modification of lipid A, via mutations on the PmrA/PmrB system causing overexpression of LPS-modified genes [28]. The resistance potential of *Salmonella* isolates to polymyxin in this study unveils both clinical and veterinary importance [29]. Not only did the NTS isolates in this study exhibit resistance potentials to several antibiotics, they equally demonstrated high pathogenicity ability to cause diseases in humans.

In fact, their belonging to large pathogenic families confirms the zoonotic status of NTS and their ability to exhibit broad-host adaptation. It is not surprising that serovar Enteritidis scored the highest probability to cause disease in humans. Previously, *Salmonella* Enteritidis has been the most prevalent world foodborne pathogen after S. Typhimurium based on its involvement in disease outbreaks [30]. Curiously, one out of two (50%) Enteritidis isolates in the current study belonged to sequence type ST-11, already incriminated as the major cause of African NTS [30]. Despite its exceptional MDR potential, serovar Poona was the least human pathogen. Its relatively low pathogenicity may reflect its low virulence power. Indeed, NTS disposes an arsenal of virulence factors to cause damage to their hosts.

The *Salmonella* pathogenicity islands described in Table 7 represent powerful virulence weapons during *Salmonella* infections. The ubiquity of C63PI may explain its role in *Salmonella* survival during iron uptake, thus its conservation among *Salmonella* species [31,32]. Out of the 11 identified SPIs, SPI-1, SPI-2, SPI-3, SPI-4, and SPI-5 play a more critical role during *Salmonella* pathogenesis [33]. While SPI-1 is mainly involved in the initiating stage of infection, SPI-2 is required for systemic infection by easing the replication of intracellular bacteria within SCV [34]. Additionally, SPI-3 is required for survival in macrophages including in a low-magnesium environment; SPI-4 is needed for intramacrophage survival, toxin secretion, and apoptosis [35]. Furthermore, *Salmonella* uses SPI-5 to induce a pro-inflammatory immune response sometimes resulting in diarrhea [36]. However, the functions of SPI-1, SPI-2, SPI-3, SPI-4, and SPI-5 could also overlap [37]. Conversely, SPI-8, SPI-9, SPI-13, and SPI-14 are associated with the regulation of the expression of other SPI genes or associated effector proteins [33] (Appendix A).

The observed variation in the SPI profile among NTS in this study unveils their differential degree of virulence. Each of the SPIs works in synergy with an arsenal of genes that code for different specialized effector proteins. These effectors are either secreted or translocated into the host cells to enable *Salmonella* to manipulate the host’s key cellular functions such as signal transduction, membrane trafficking, and immune responses. In occurrence virulence factors, notably SipA, SipC, SopB, SopD, SopE, SopE2, SipB, EnvE, EnvF, InvA, InvE, SipD, Sii E, IagB, SptP, OrgB, PhoP, MisL, ShdA, and rtn detected in roughly 82% of *Salmonella* isolates appear necessary to facilitate their entry into host cells to establish infection (Appendix A). Interestingly, all the effectors described in the *Salmonella* entry process are SPI-1 encoded [38,39]. Putative fimbriae usher (FimA, FimC, FimZ, HilD), pilin chaperon (ecpD1, CsgD), and flagellin (proteins FliZ, FliT) collectively called adhesion factors are additional virulence determinants to ensure successful colonization and persistence of *Salmonella* in the hosts [32]. Moreover, the expression of “SipA and SipC” and “SopE and SopE2” required, respectively, for invasion and internalization efficiency [38,39] reinforced the pathogenicity potentials of our isolates. The expression of SopB, SopE, SopA, SopD, SpiC, SpvB, SseB, SseC, SseD, PipB2, SptP, SopE2, and antigen presentation proteins, especially SpaK, SpaN, SpaS, and SpaR was sufficient for the maturation and trafficking process of SCV, replication within SCV and the monitoring of the host immune responses. Likewise, virulence determinants such as heat shock protein, stress protein, CuSA, ScsC, and putative ABC-type bacteriocin exporter could enable isolates to successfully control competing bacteria during infection [37,40,41].

Though HilA, the leading regulator of SPI-1 [38] was not detected in the isolates, between 59% and 68% of NTS isolates possessed PhoP, HilD, and FliZ, three important transcriptional regulators; also, 95% carried H-NS, the master silencer of horizontal transfer genes [37,42] to regulate the secretion or translocation of effector proteins during each step of *Salmonella* infection [43]. Otherwise, virulence plasmids were not widely distributed among NTS in this study (Appendix A). IncF plasmid known as a crucial virulence plasmid [19] was carried by *Salmonella* isolates that harbored the gene *qnrB*. This observation corroborates a previous study that seemed to establish a certain relationship between harboring IncF plasmid and fluoroquinolone-resistance potential [44]. Despite the wide distribution of IncI1 plasmids in *Salmonella*, [18,45] only 23% of our NTS isolates harbored them. The restricted presence of Col (PHAD28) plasmid to MDR Poona isolates seems to underline a certain correlation between harboring this plasmid and the MDR potential of the host [18]. Equally, the exclusive presence of IncH plasmid seems to correlate well with the detection of genes *cat*, *qnr*, *strAB*, *TEM-1*, and *tet* in isolate 34de as previously reported [18]. Remarkably, all the aforementioned plasmids are mobilizable plasmids that are known to promote multiple drug resistance in bacteria [18].

## 4. Conclusions

Although the fact that the samples analyzed were limited to a single abattoir, this study has revealed the public health importance of *Salmonella* isolates at the Yaounde abattoir. Despite a relatively moderate phenotypic resistance to streptomycin, genotypically, *Salmonella* isolates possessed high MDR potentials against several classes of antibiotics notably third-generation cephalosporin and fluoroquinolone. More than 31% of the isolates exhibited resistance potential to polymyxin, considered a critically important drug. Additionally, close to 80% of NTS isolates harbored “silent genes” which could act as a reservoir of resistance to foodborne bacteria. The role of plasmids should be integrated into the antibiotic surveillance program. *Salmonella* isolates also exhibited a high degree of pathogenicity and virulence to establish infection in their hosts including humans. The presence of virulence determinants is crucial in *Salmonella* pathogenicity and in understanding the epidemiological knowledge about the potential severity of infections and to mitigate potential outbreaks of Salmonellosis. The combined effect of high pathogenicity and MDR potentials of NTS at the Yaounde abattoir highlights the need for improvement in food safety practices and the need for antibiotic stewardship in livestock production systems in Cameroon. Given the fact that there is a lack of reliable data on the NTS AMR prevalence in Cameroon, this study suggests that further studies are needed to elucidate the epidemiology of the antibiotic resistance among *Salmonella* across all the Cameroonian slaughterhouses including characterizing the genes involved and the plasmids that carry them.

## 5. Materials and Methods

### 5.1. Bacterial Isolates

This study was a cross-sectional study. Prior to slaughter, five cattle were randomly chosen per week for every sampling session following the Meat Industry Guide describing sampling frequency for red meat carcasses [46]. Moreover, five butchers (one-fifth of the butchers at duty) were recruited for every sampling session. Wet swabs (n = 830) from live cattle (n = 145), beef carcasses (n = 435), butchers’ hands (n = 145) and the meat contact surfaces (n = 105), were aseptically collected between December 2014 and November 2015 at the Yaounde abattoir to isolate non-typhoidal *Salmonella* (NTS) following ISO 6579 [47]. Finally, 23 NTS were isolated and were distributed as follows: live cattle (n = 1), beef carcasses (n = 19), processing environments (n = 2), and butchers’ hands (n = 1) All *Salmonella* isolates were confirmed using API-20 E kit (BioMérieux, France) and a qualitative real-time PCR assay [48].

### 5.2. Antibiotic Susceptibility Tests

*Salmonella* concentration (1.5 × 10^8^ CFU/mL) and those of controls (*Escherichia coli* ATCC 25922 and *Staphylococcus aureus* ATCC 43300) were spread onto the surface of Mueller–Hinton agar to which the antibiotic disks were placed and incubated for 18 to 24 h. The diameter of the zones of inhibition around each antibiotic disk was measured with a ruler and recorded to the nearest millimeter and isolates were classified as resistant, susceptible, or intermediate [49]. The antibiotics used were ampicillin (AMP) 10 μg, chloramphenicol (C) 30 μg, tetracycline (TE) 30 μg, and streptomycin (STR) 25 μg.

### 5.3. DNA Extraction

Total DNA was extracted from *Salmonella* overnight culture using Quick-DNA™ Miniprep Plus Kit (Zymo Research, Irvine, CA, USA) following the manufacturer’s instructions. The purified DNA was quantified using a NanoDrop 2000c spectrophotometer (ThermoFisher Scientific, Santa Clara, CA, USA) and stored at −20 °C until use.

### 5.4. Library Preparation and Sequencing

Paired-end libraries were constructed with 0.2 ng/µL of purified DNA using the Nextera XT DNA Library Prep Kit as recommended by the manufacturer (Illumina, San Diego, CA, USA) and were quantified using a Qubit fluorometer (ThermoFisher Scientific, USA). WGS was performed on Illumina NextSeq platform using NextSeq 500/550 high-output kit v2 (300 cycles) at Murdoch University (Australia) and on Illumina Miseq platform using pair-ended MiSeq reagent v3 kit (2 × 201 bp) at the BecA-ILRI Hub (Nairobi, Kenya) following the manufacturers’ guidelines.

### 5.5. Resistance Genes, Pathogenicity, and Virulence Factors

The qualities of the raw sequences were checked with FASTQC and trimmed using Trimmomatic 2.6 at Q score below 20. The trimmed data were assembled using SPAdes version 3.11, and genomes were annotated with the NCBI Prokaryotic Genome Annotation Pipeline [50]. The antibiotic resistance genes (ARG) in the assembled genomes were identified by BLAST search against the reference ARG sequences from ResFinder 3.0 [23] and CARD 2017 [51] with ≥95% gene identity and 60% sequence length of the resistance gene. The pathogenicity of NTS was predicted using PathogenFinder 1.1 [52] with the threshold for minimum % identity at 95% and minimum % coverage at 60%. The *Salmonella* pathogenicity islands were detected using SPIFinder 1.0 [33] at ≥95% gene identity and 60% sequence length cut off. Acquired virulence genes were detected by uploading the raw reads into VirulenceFinder 2.0 [53] while virulence plasmid was assessed using PlasmidFinder 2.1 [53] with the threshold for % identity at minimum % coverage at 60%.

### 5.6. Statistical Analysis

Statistical significance for all tests was set at the level of *p* ≤ 0.05, using the methods of Duncan [54], and descriptive statistics were calculated for all variables as appropriate using IBM SPSS 20 [55]. Sensitivity and specificity were calculated to determine the relationship between phenotypic and genotypic profiles of NTS following the method described by Genders et al. [56]. The odds ratio was determined using online MedCalc Version 20.027.

## Figures and Tables

**Table 1 pathogens-11-00502-t001:** Antimicrobial sensitivity of non-typhoidal *Salmonella* isolated at the Yaounde abattoir.

Sample Code	*Salmonella* Isolate	Tetracycline	Chloramphenicol	Streptomycin	Ampicillin
34ev	Wernigerode	**−**	**−**	**−**	**−**
8ev	Poona	**+**	**+**	**+**	**+**
35dea	Wilhelmsburg	**−**	**−**	**−**	**−**
35deb	Wilhelmsburg	**−**	**−**	**−**	**−**
22sa	Poona	**−**	**+**	**+**	**+**
31eva	Wilhelmsburg	**−**	**−**	**−**	**−**
31evb	Wilhelmsburg	**−**	**−**	**−**	**−**
32eva	Wilhelmsburg	**−**	**−**	**−**	**−**
32evb	Wernigerode	**−**	**−**	**−**	**−**
86ev	Infantis	**−**	**−**	**+**	**I**
100ev	Wernigerode	**−**	**−**	**−**	**−**
36ev	Wilhelmsburg	**−**	**−**	**−**	**−**
98se	Wernigerode	**−**	**−**	**−**	**−**
108ev	Kibusi	**−**	**−**	**−**	**−**
20de	Enteritidis	**−**	**−**	**+**	**−**
34de	Poona	**+**	**+**	**+**	**+**
60sa	Enteritidis	**−**	**−**	**I**	**−**
88sa	Infantis	**−**	**−**	**−**	**−**
88sab	Infantis	**−**	**−**	**−**	**−**
133sa	Enteritidis	**−**	**−**	**−**	**I**
103bo	Wilhelmsburg	**−**	**−**	**−**	**−**
EVJUL	Mbandaka	**−**		**+**	**−**
DEF1	Not sequenced	**−**	**−**	**−**	**−**

NB: **+** indicates resistance to the test antibiotic; **I** = intermediate isolates; **−** susceptible isolates; code and serovar of *Salmonella* strain written in red, represent the multidrug-resistant (MDR) strains.

**Table 2 pathogens-11-00502-t002:** Distribution of resistance genes against specific antibiotic across *Salmonella* isolates in the study.

Antibiotics	Resistance Genes	8ev	20de	22sa	31eva	31evb	32eva	32evb	34de	34ev	35dea	35deb	36ev	60sa	88sa	88sab	98se	100ev	103bo	108ev
		Poona	Enteritidis	Poona	Wilhelmsburg	Wilhelmsburg	Wilhelmsburg	wernigerode	Poona	Wilhelmsburg	Wilhelmsburg	Wilhelmsburg	Wilhelmsburg	Enteritidis	Infantis	Infantis	wernigerode	wernigerode	Wilhelmsburg	Kibusi
STR	*aadA*	**−**	**+**	**−**	** + **	**−**	**−**	**−**	**−**	**−**	** + **	**−**	**−**	**−**	**−**	**−**	**−**	**−**	**−**	**−**
	*aadA1*	**+**	**+**	**+**	**−**	**−**	**−**	**−**	**+**	**−**	**−**	**−**	**−**	** + **	**−**	**−**	**−**	**−**	**−**	**−**
	*aadA2*	**+**	**−**	**−**	** + **	**−**	**−**	**−**	**+**	**−**	**−**	**−**	**−**	**−**	** + **	**−**	**−**	**−**	**−**	**−**
CHL	*Cat*	**−**	** + **	**+**	**−**	**−**	**−**	** + **	**−**	**−**	**−**	**−**	**−**	**−**	**−**	**−**	**−**	**−**	**−**	** + **
	*catI*	**+**	** + **	**−**	**−**	**−**	**−**	**−**	**+**	**−**	**−**	**−**	**−**	**−**	**−**	**−**	**−**	**−**	**−**	**−**
	*CatA2&3*	**−**	**−**	**−**	** + **	**−**	**−**	**−**	**−**	**−**	**−**	**−**	**−**	** + **	**−**	**−**	**−**	**−**	**−**	**−**
	*cmlA1,5,6*	** + **	** + **	**−**	**−**	**−**	**−**	**−**	** + **	**−**	**−**	**−**	**−**	**−**	**−**	**−**	**−**	**−**	**−**	**−**
TE	*tet(A)*	**+**	**−**	**−**	**−**	**−**	**−**	**−**	**−**	**−**	**−**	**−**	**−**	**−**	**−**	**−**	**−**	**−**	**−**	**−**
	*tet(B), (C)*	**−**	**−**	** + **	** + **	**−**	**−**	** + **	** + **	**−**	**−**	**−**	**−**	**−**	** + **	**−**	**−**	**−**	**−**	**−**
	*tet(G)*	**+**	**−**	** + **	** + **	**−**	**−**	** + **	** + **	**−**	**−**	**−**	**−**	**−**	** + **	**−**	**−**	**−**	**−**	**−**
	*tetR*	**+**	** + **	**−**	** + **	** + **	** + **	**−**	** + **	** + **	** + **	**−**	** + **	** + **	** + **	** + **	** + **	** + **	** + **	** + **
AMP	*TEM-1*	** + **	** + **	**−**	**−**	**−**	**−**	**−**	**−**	**−**	**−**	**−**	**−**	** + **	**−**	**−**	**−**	**−**	**−**	**−**
	*TEM-163*	**+**	** + **	**−**	**−**	**−**	**−**	**−**	**+**	**−**	**−**	**−**	**−**	**−**	** + **	**−**	**−**	**−**	**−**	**−**

**+**, detected presence of resistance gene; **−**
, resistance gene not detected; **+**, false negative; AMP = Ampicillin; CHL = Chloramphenicol; STR = Streptomycin; TE = Tetracycline.

**Table 3 pathogens-11-00502-t003:** Matching antibiotic susceptibility test with genotypic resistance potentials of *Salmonella* isolates.

Antibiotics	Number of Test Results						Sensitivity (%)	Specificity (%)	Negative PV (%)	Positive PV (%)
	Resistant Phenotype Sensitive Phenotype									
	Genotype (TP)	Genotype (FP)	Genotype (FN)	Genotype (TN)	Odds Ratio	*p* Value				
STR	5	0	3	11	4.0	0.46	62.5	100	78.6	100
AMP	3	0	3	13			50	100	81.2	100
CHL	3	0	5	11			62.5	100	68.7	100
TE	3	0	15	1			16.7	100	6.2	100

Key: Sensitivity = TP/(TP + FN) × 100; Specificity = TN/(FP + TN) × 100; PPV = TP/(TP + FP) × 100; NPV = TN/(FN + TN) × 100; TP = True positive; FP = False positive; TN = True negative; FN = False negative; PPV = Positive predictive value; NPV = Negative predictive value.

**Table 4 pathogens-11-00502-t004:** Distribution of multidrug resistance genes among *Salmonella* serotypes.

Class of Antibiotic	Resistance Gene	8ev	20de	22sa	31eva	31evb	32eva	32evb	34de	34ev	35dea	35deb	36ev	60sa	88sa	88sab	98se	100ev	103bo	108ev
		Poona	Enteritidis	Poona	Wilhelmsburg	Wilhelmsburg	Wilhelmsburg	Wernigerode	Poona	Wilhelmsburg	Wilhelmsburg	Wilhelmsburg	Wilhelmsburg	Enteritidis	Infantis	Infantis	Wernigerode	Wernigerode	Wilhelmsburg	Kibusi
Aminoglycosides	* acrD *	**+**	**−**	**−**	**−**	**−**	**−**	**−**	**−**	**−**	**−**	**−**	**−**	**−**	**−**	**−**	**−**	**−**	**−**	**−**
	*bacA*	**+**	**−**	**−**	**−**	**−**	**−**	**−**	**+**	**−**	**−**	**−**	**−**	**−**	**−**	**−**	**−**	**−**	**−**	**−**
	* baeR *	**+**	**−**	**−**	**+**	**−**	**−**	**+**	**+**	**+**	**+**	**+**	**+**	**+**	**−**	**−**	**+**	**+**	**+**	**+**
	* cpxA *	**+**	**+**	**+**	**−**	**−**	**−**	**+**	**+**	**+**	**+**	**+**	**+**	**+**	**+**	**+**	**+**	**−**	**−**	**−**
Phenicol	* acrB *	**−**	**−**	**−**	**−**	**−**	**−**	**−**	**+**	**−**	**−**	**−**	**−**	**−**	**−**	**−**	**−**	**−**	**−**	**−**
	* golS *	**+**	**+**	**+**	**+**	**+**	**+**	**+**	**+**	**+**	**+**	**+**	**+**	**+**	**+**	**+**	**+**	**+**	**+**	**+**
	* mdsABC *	**+**	**+**	**+**	**+**	**+**	**+**	**+**	**+**	**+**	**+**	**+**	**+**	**+**	**+**	**+**	**+**	**+**	**+**	**+**
	* sdiA *	**+**	**+**	**+**	**+**	**+**	**+**	**+**	**+**	**+**	**+**	**+**	**+**	**+**	**+**	**+**	**+**	**+**	**+**	**+**
	* E.coli soxR&S *	**+**	**+**	**+**	**+**	**+**	**+**	**+**	**+**	**+**	**+**	**+**	**+**	**+**	**+**	**+**	**+**	**+**	**+**	**+**
	* ramR *	**−**	**−**	**−**	**−**	**−**	**−**	**−**	**−**	**−**	**−**	**−**	**−**	**−**	**−**	**−**	**−**	**+**	**−**	**−**
	* TolC *	**+**	**+**	**+**	**+**	**+**	**+**	**+**	**+**	**+**	**+**	**+**	**+**	**+**	**+**	**+**	**+**	**+**	**+**	**+**
Tetracycline	* acrB *	**−**	**−**	**+**	**−**	**−**	**−**	**−**	**−**	**−**	**−**	**−**	**−**	**−**	**−**	**−**	**−**	**−**	**−**	**−**
	* acrD *	**+**	**−**	**−**	**+**	**−**	**−**	**−**	**+**	**−**	**−**	**−**	**−**	**−**	**−**	**−**	**−**	**+**	**−**	**−**
	* golS *	**−**	**−**	**−**	**−**	**−**	**−**	**−**	**+**	**−**	**−**	**−**	**−**	**−**	**−**	**−**	**−**	**−**	**−**	**−**
	* mdfA *	**−**	**−**	**−**	**−**	**−**	**−**	**−**	**+**	**−**	**−**	**−**	**−**	**−**	**−**	**−**	**−**	**−**	**−**	**−**
	* ramA *	**−**	**−**	**−**	**−**	**−**	**−**	**−**	**−**	**−**	**−**	**−**	**−**	**−**	**−**	**−**	**−**	**+**	**−**	**−**
	* sdiA *	**+**	**+**	**+**	**+**	**+**	**+**	**+**	**+**	**+**	**+**	**+**	**+**	**+**	**+**	**+**	**+**	**+**	**+**	**+**
	* E.col soxR&S *	**+**	**−**	**+**	**−**	**−**	**−**	**+**	**+**	**+**	**+**	**+**	**+**	**+**	**+**	**+**	**+**	**+**	**+**	**+**
β-lactam	* acrB *	**−**	**−**	**+**	**−**	**−**	**−**	**−**	**+**	**−**	**−**	**−**	**−**	**−**	**−**	**−**	**−**	**−**	**−**	**−**
	*CMY-9*	**−**	**−**	**−**	**−**	**−**	**−**	**−**	**+**	**−**	**−**	**−**	**−**	**−**	**−**	**−**	**−**	**−**	**−**	**−**
	* CTX-M-14 *	**+**	**−**	**−**	**−**	**−**	**−**	**−**	**+**	**−**	**−**	**−**	**−**	**−**	**−**	**−**	**−**	**−**	**−**	**−**
	*DHA-1*	**−**	**−**	**−**	**−**	**−**	**−**	**−**	**+**	**−**	**−**	**−**	**−**	**−**	**−**	**−**	**−**	**−**	**−**	**−**
	*FOX-1*	**+**	**+**	**+**	**+**	**+**	**+**	**+**	**+**	**+**	**+**	**+**	**+**	**+**	**+**	**+**	**+**	**+**	**+**	**+**
	* golS *	**+**	**+**	**+**	**−**	**−**	**−**	**+**	**+**	**−**	**−**	**−**	**+**	**−**	**−**	**−**	**−**	**−**	**−**	**−**
	* marA *	**−**	**+**	**−**	**+**	**+**	**−**	**+**	**−**	**−**	**−**	**−**	**+**	**−**	**−**	**+**	**+**	**−**	**+**	**+**
	* OXA-1 *	**+**	**−**	**+**	**−**	**−**	**−**	**−**	**+**	**−**	**−**	**−**	**−**	**−**	**−**	**−**	**−**	**−**	**−**	**−**
	* OXA-2 *	**+**	**−**	**−**	**−**	**−**	**−**	**−**	**+**	**−**	**−**	**−**	**−**	**−**	**−**	**−**	**−**	**−**	**−**	**−**
	* OXA-7 *	**−**	**−**	**−**	**−**	**−**	**−**	**−**	**+**	**−**	**−**	**−**	**−**	**−**	**−**	**−**	**−**	**−**	**−**	**−**
	* ramR *	**−**	**−**	**−**	**−**	**−**	**−**	**−**	**−**	**−**	**−**	**−**	**−**	**−**	**−**	**−**	**−**	**+**	**−**	**−**
	* E.coli soxR&S *	**+**	**−**	**−**	**+**	**+**	**+**	**+**	**+**		**+**	**+**	**+**	**+**	**+**	**−**	**+**	**+**	**+**	**+**
Fluoroquinolone	* acrB *	**−**	**−**	**−**	**−**	**−**	**−**	**−**	**+**	**−**	**−**	**−**	**−**	**−**	**−**	**−**	**−**	**−**	**−**	**−**
	* acrF *	**−**	**−**	**−**	**−**	**−**	**−**	**−**	**−**	**−**	**−**	**−**	**−**	**−**	**−**	**−**	**−**	**−**	**−**	**−**
	* CRP *	**−**	**−**	**−**	**−**	**−**	**−**	**−**	**+**	**+**	**+**	**+**	**+**	**+**	**+**	**−**	**+**	**+**	**+**	**−**
	* E.coli soxR&S *	**+**	**+**	**+**	**+**	**+**	**+**	**+**	**+**	**+**	**+**	**+**	**+**	**+**	**+**	**+**	**+**	**+**	**+**	**+**
	* emrB *	**+**	**−**	**−**	**−**	**−**	**−**	**−**	**+**	**−**	**−**	**−**	**−**	**−**	**−**	**−**	**+**	**+**	**+**	
	* mdtK *	**+**	**+**	**+**	**+**	**+**	**+**	**+**	**+**	**+**	**+**	**+**	**+**	**+**	**+**	**+**	**+**	**+**	**+**	**+**
	* patA *	**−**	**−**	**+**	**−**	**−**	**−**	**−**	**−**	**−**	**−**	**−**	**−**	**−**	**−**	**−**	**−**	**−**	**−**	**−**
	* qnrB1 *	**+**	**+**		**−**	**−**	**−**	**−**	**+**	**−**	**−**	**−**	**−**	**+**	**−**	**−**	**−**	**−**	**−**	**−**
	* ramR *	**−**	**−**	**−**	**−**	**−**	**−**	**−**	**+**	**−**	**−**	**−**	**−**	**−**	**−**	**−**	**−**	**−**	**−**	**−**
	* sdiA *	**+**	**+**	**+**	**+**	**+**	**+**	**+**	**+**	**+**	**+**	**+**	**+**	**+**	**+**	**+**	**+**	**+**	**+**	**+**
	* TolC *	**+**	**−**	**+**	**−**	**−**	**−**	**−**	**+**	**−**	**−**	**−**	**−**	**−**	**−**	**−**	**−**	**−**	**−**	**−**
Sulfa	*Sul1*	**+**	**+**	**−**	**−**	**−**	**−**	**−**	**−**	**−**	**−**	**−**	**−**	**+**	**−**	**−**	**−**	**−**	**−**	**−**
	*Sul2*	**+**	**−**	**−**	**−**	**−**	**−**	**−**	**−**	**−**	**−**	**−**	**−**	**−**	**−**	**−**	**−**	**−**	**−**	**−**
Macro lide	* TolC *	**+**	**−**	**+**	**−**	**−**	**−**	**−**	**+**	**−**	**−**	**−**	**−**	**−**	**−**	**−**	**−**	**−**	**−**	**−**
	*CRP-7*	**+**	**+**	**−**	**−**	**−**	**−**	**−**	**+**	**+**	**+**	**+**	**+**	**+**	**+**	**−**	**+**	**+**	**+**	**−**
	*macA*	**+**	**+**	**+**	**+**	**−**	**+**	**+**	**+**	**−**	**−**	**−**	**+**	**−**	**−**	**+**	**−**	**+**	**−**	**−**
PMB	* mgrB *	**+**	**+**	**+**	**+**	**+**	**+**	**+**	**+**	**−**	**+**	**−**	**+**	**+**	**+**	**−**	**+**	**+**	**+**	**+**

**+** = Presence multiple resistance gene; **−** = no multiple resistance gene detected; Sulfa = Sulfonamide; PMB = Polymyxin B; Genes in purple = multiple resistance genes; genes in red = genes coding resistance against antibiotics high concern by WHO; Genes in black = represent genes that code resistance against not more than one class of antibiotics.

**Table 5 pathogens-11-00502-t005:** Polymorphism in *Salmonella* serovars with potential resistance to polymyxins.

Isolate Code	Position of Mutation on pmrA & pmrB	Nucleotide Change	Amino Acid Change
8ev	pmrB		
	pmrB p.M15T	ATG → ACT	M → T
	pmrB p.G73S	GGC → AGC	G → S
	pmrB p.V74I	GTA → ATA	V → I
	pmrA		
	pmrA p.T89S	ACC → AGC	T → S
22sa	pmrB		
	pmrB p.M15T	ATG → ACT	M → T
	pmrB p.G73S	GC → AGC	G → S
	pmrB p.A111T	GCG → ACG	A→ T
	pmrA		
	pmrA p.T89S	ACC → AGC	T → S
31eva	pmrB		
	pmrB p.M15T	ATG→ACT	M → T
	pmrB p.G73S	GGC → AGC	G → S
	pmrB p.V74I	GTA → ATA	V →I
	pmrB p.A111T	GCG →ACG	A → T
	pmrA p.T89S	ACC → AGC	T → S
32eva	pmrB		
	pmrB p.I18L	ATT→CTT	I → L
34de	pmrB		
	pmrB p.M15T	ATG → ACT	M → T
	pmrB p.G73S	GGC → AGC	G → S
	pmrB p.V74I	GTA → ATA	V → I
	pmrB p.A111T	GCG → ACG	A → T
	pmrB p.L352M	CTG → ATG	L → M
	pmrA		
	pmrA p.T89S	ACC → AGC	T → S
35dea	pmrB		
	pmrB p.V126G	GTC → GGC	V → G
	pmrB p.S127A	TCG → GCG	S → A
	pmrB p.I129L	ATC → CTC	I → L
	pmrB p.V133D	GTT → GAT	V → D
	pmrB p.L136R	TTG → AGG	L → R
	pmrB p.T139P	ACG → CCG	T → P

**Table 6 pathogens-11-00502-t006:** Prediction of *Salmonella* isolates as human pathogens.

Serovars	PHPathogen	Proteome Coverage (%)	Matched PF	Non PF
Poona	0.93 ^a^	10.1	466	5
Enteritidis	0.95 ^a^	18.2	787	2
Wilhelmsburg	0.94 ^a^	15.71	691	3
Wernigerode	0.94 ^a^	15	661	3
Infantis	0.94 ^a^	17.45	746	3
Kibusi	0.94 ^a^	18	778	3

Key: PHPathogen = predicted as human pathogen; PF = pathogenic family; NPF = Non pathogenic family. Values with the same superscript letters are not significantly different (*p* > 0.05).

**Table 7 pathogens-11-00502-t007:** Virulence factors among *Salmonella* serovars.

*Salmonella* Pathogenicity Island (SPI)	Function Category	8ev	20de	22sa	31eva	31evb	32eva	32evb	34de	34ev	35dea	35deb	36ev	60sa	88sa	88sab	98se	100ev	103bo	108ev
Gene Locus		Poona	Enteritidis	Poona	Wilhelmsburg	Wilhelmsburg	Wilhelmsburg	Wernigerode	Poona	Wilhelmsburg	Wilhelmsburg	Wilhelmsburg	Wilhelmsburg	Enteritidis	Infantis	Infantis	Wernigerode	Wernigerode	Wilhelmsburg	Kibusi
SPI-1	5	**−**	**−**	**−**	**−**	**−**	**−**	**−**	**−**	**−**	**+**	**−**	**+**	**−**	**−**	**−**	**+**	**−**	**−**	**−**
SPI-2	14	**−**	**−**	**−**	**−**	**−**	**−**	**−**	**+**	**−**	**−**	**+**	**+**	**+**	**+**	**+**	**+**	**−**	**−**	**−**
SPI-3	15	**−**	**−**	**−**	**+**	**−**	**+**	**−**	**−**	**−**	**+**	**+**	**−**	**−**	**+**	**+**	**+**	**−**	**−**	**−**
SPI-3	16	**−**	**−**	**−**	**+**	**+**	**−**	**+**	**−**	**+**	**−**	**−**	**+**	**+**	**+**	**−**	**−**	**+**	**−**	**−**
SPI-3	15	**−**	**−**	**−**	**−**	**−**	**−**	**−**	**−**	**−**	**+**	**−**	**−**	**+**	**−**	**−**	**−**	**−**	**−**	**−**
SPI-4	17	**−**	**−**	**−**	**−**	**−**	**−**	**−**	**−**	**−**	**−**	**+**	**+**	**−**	**−**	**+**	**−**	**−**	**−**	**−**
SPI-5	18	**−**	**+**	**−**	**−**	**−**	**−**	**−**	**−**	**−**	**+**	**−**	**−**	**+**	**+**	**−**	**−**	**−**	**+**	**−**
SPI-8	21	**−**	**−**	**−**	**+**	**+**	**+**	**+**	**−**	**+**	**−**	**+**	**+**	**−**	**−**	**−**	**−**	**−**	**−**	**−**
SPI-9	22	**−**	**−**	**−**	**−**	**−**	**−**	**−**	**−**	**−**	**−**	**−**	**−**	**+**	**+**	**−**	**−**	**−**	**−**	**−**
SPI-13	9	**+**	**+**	**+**	**−**	**−**	**−**	**−**	**+**	**−**	**−**	**−**	**−**	**+**	**+**	**−**	**−**	**−**	**+**	**+**
SPI-13	10	**+**	**+**	**+**	**−**	**−**	**−**	**−**	**+**	**−**	**−**	**−**	**−**	**+**	**+**	**−**	**−**	**−**	**+**	**+**
SPI-13	11	**+**	**+**	**+**	**−**	**−**	**−**	**−**	**+**	**−**	**−**	**−**	**−**	**+**	**+**	**−**	**−**	**−**	**+**	**+**
SPI-14	12	**+**	**+**	**+**	**−**	**−**	**−**	**−**	**+**	**−**	**−**	**−**	**−**	**+**	**+**	**−**	**−**	**−**	**+**	**+**
SPI-14	13	**+**	**−**	**+**	**−**	**−**	**−**	**−**	**+**	**−**	**+**	**−**	**−**	**+**	**+**	**−**	**−**	**−**	**+**	**+**
C63PI	1	**+**	**+**	**−**	**+**	**+**	**+**	**+**	**+**	**+**	**+**	**+**	**+**	**+**	**+**	**+**	**+**	**+**	**+**	**+**

**+** = detected presence of virulence factor; **−** = Virulence factor not detected.

## Data Availability

The data that support the findings of this study are available from the corresponding author upon reasonable request.

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
