# Peer review of "Detection of Antimicrobial Resistance, Pathogenicity, and Virulence Potentials of Non-Typhoidal Salmonella Isolates at the Yaounde Abattoir Using Whole-Genome Sequencing Technique"

_pathogens, 2022, doi:10.3390/pathogens11050502_

Round 1
Reviewer 1 Report
This study was aimed at predicting MDR, pathogenicity, and virulence
potentials of Salmonella isolated at the Yaounde abattoir using WGS
Major concerns
1- However the study is interesting but the number of collected samples is very low
2- The introduction needs a lot of improvement. For example, you did not talk about your problem and its economic importance.
3- In table 1: you can put the size of the inhibition zone in the supplementary material. It's better to provide data about resistance and susceptibility
4- The name of the bacteria needs to be in italic.
5- Tables 7, 8a, 8b: needs to move as supplementary, you can use this data in the discussion as this is not a review article
6- remove the subtitles from the discussion and rework the discussion again.
7- Please provide a correlation between genotypic and phenotypic data as shown in this study. You can cite it or other papers in the analysis methodology part if you like https://www.mdpi.com/2079-6382/10/12/1450
8- You need to add a section about the used statistical analysis you used. I did not see any stats throughout your manuscript.
Author Response
Reviewers’ comments#1: However the study is interesting but the number of collected samples is very low
Response#1: The number of samples (n=830 wet swabs) that was mistaken for 23 NTS isolates is highlighted in the abstract (see line 26) and in the materials and methods section (see lines 358-362) in the revised manuscript.
Observation#1: The worry concerning the sample size has been addressed.
Reviewers’ comments#2: The introduction needs a lot of improvement. For example, you did not talk about your problem and its economic importance.
Response#2: The introduction has been great improved by showing the public health and economic importance of NTS in Africa and in Cameroon in particular (see lines 50-58). The context of the Yaounde abattoir is highlighted (see lines 68-74) in the revised manuscript.
Observation#2: Thanks for giving me the opportunity to improve on the quality of the introduction
Reviewers’ comments#3: In table 1: you can put the size of the inhibition zone in the supplementary material. It's better to provide data about resistance and susceptibility.
Response#3: Table 1 has been modified (see line 90 in the revised manuscript) and the size of the inhibition zones are found in the supplementary material (see Table S2).
Observation#3: Thanks
Reviewers’ comments#4: The name of the bacteria needs to be in italic
Response#4: The name of bacteria has been italicized accordingly across the revised manuscript (see lines24,28-29, 38,64, 209, 225,280, 305, 307, 377, 484, 486-487, 489, 499, 504-505, 510, 515, 518, 521, 528, 533, 535, 540, 542-543, 545, 549, 551, 564.
Observation#4: Thanks
Reviewers’ comments#5: Tables 7, 8a, 8b: needs to move as supplementary, you can use this data in the discussion as this is not a review article
Response#5: Tables 7, 8a and 8b have been moved to the supplementary material (see Tables S4, S5a and 5b, respectively) and reported in the results section (see lines 193, 195) and in the discussion section (See lines 317, 326, 345) of the revised manuscript
Observation#5: Thanks
Reviewers’ comments#6: Remove the subtitles from the discussion and rework the discussion again.
Response#6: Subtitles have been removed from the discussion and the different paragraphs have been networked (see discussion section of the revised manuscript)
Observation#6: Thanks
Reviewers’ comments#7: Please provide a correlation between genotypic and phenotypic data as shown in this study. You can cite it or other papers in the analysis methodology part if you like https://www.mdpi.com/2079-6382/10/12/1450
Response#7: Correlation between phenotypic and genotypic resistance of NTS has been inserted in the results section (see lines113-127) and in the discussion (see lines 244-255)
Observation#7: Thanks
Reviewers’ comments#8: You need to add a section about the used statistical analysis you used. I did not see any stats throughout your manuscript
Response#8: Statistical analysis has been added to the materials and methods section (see lines 406-412)
Observation#8: Thanks
Reviewer 2 Report
Dear authors, I read your paper about the detection of antimicrobial resistance, pathogenicity and virulence potentials on NTS in Cameroon.
I am afraid to say that you guys have a case report of 23 isolates instead a scientific paper in hands. In this sense, I recommended major review. On possibility could be a short communication.
I suggest to the authors to make some changes in the manuscript regardless the editor's decision.
General:
Please, be consistent with Salmonella sp. nomenclature. Sometimes they are italicized, sometimes not.
Please, try to clarify the relevance of these findings for the construction of scientific knowledge about Salmonella sp. AMR in your country.
In the introduction:
Can you explore a bit more this Yamounde abattoir? Is it the main slaughterhouse in the country? Is it the only one? Please, give to the reader a bit of context.
Lines 50-53: This sentence is not correct. Animals are not the mais source for AMR. They can have a certain importance, but are not the major source.
In the M&M:
Why did you collected 23 samples in 1 year? I wonder how relevant is to use 1 isolate from cattle; 1 hands; 2 environment... Can you give more information about the reasoning behind the sample strategy?
I am aware of the fact that most of the softwares for bioinformatics are "black boxes", but can you explain a bit each analysis you did? For instance, does PathogenFinder 1.1 use an artificial intelligence algorithm? The same for other algorithms used here.
In the results:
Lines 79-85: Be consistent with the number of decimals. Also be consistent with the use of % symbol.
Line 95: Not sure if you really meant de-tected or detected.
Line 132: Not sure if you really meant pre-dicted or predicted.
In the discussion:
Lines 169-171: Not the best way to put that. your findings highlight the need for a deep knowledge on the relevance of food chain on Salmonella sp. AMR. Further studies include in-depth surveys, longitudinal studies and risk assessment models.
Lines 194- 200: Be consistent with the genes names, sometimes they are italicized, sometimes not.
Lines 228: there is an extra space (enter) there
Lines 239: 50% of 2 is 1, correct? You should probably mention it.
Lines 241-234: Not sure the meaning of this sentence.
In the conclusion:
Lines 346-349: I do not agree with this sentence. This study calls for more information about the potential of the food chain in AMR spread in Cameroon.
Author Response
Reviewers’ comments: I am afraid to say that you guys have a case report of 23 isolates instead a scientific paper in hands. In this sense, I recommended major review. On possibility could be a short communication.
I suggest to the authors to make some changes in the manuscript regardless the editor's decision
Response: The sample size (n=830 wet swabs) that was mistaken for 23 NTS isolates is highlighted in the abstract (see line 26) and in the materials and methods section (see lines 358-362) in the revised manuscript
General: Please, be consistent with Salmonella sp. nomenclature. Sometimes they are italicized, sometimes not.
Response: The name of Salmonella has been italicized accordingly across the revised manuscript (see lines24,28-29, 38,64, 209, 225,280, 305, 307, 377, 484, 486-487, 489, 499, 504-505, 510, 515, 518, 521, 528, 533, 535, 540, 542-543, 545, 549, 551, 564
Observation: Thanks
General: Please, try to clarify the relevance of these findings for the construction of scientific knowledge about Salmonella sp. AMR in your country
Response: The development and spread of AMR among NTS in an environmental setting like the Yaounde abattoir highlights the need for the implementation of antibiotic stewardship in livestock production systems in Cameroon (see lines 220-223, 424-429)
Observation: Thanks
In the introduction: Can you explore a bit more this Yamounde abattoir? Is it the main slaughterhouse in the country? Is it the only one? Please, give to the reader a bit of context.
Response: The context of the Yaounde abattoir is highlighted (see lines 68-74) in the revised manuscript.
Observation: Thanks
In the introduction: Lines 50-53: This sentence is not correct. Animals are not the mais source for AMR. They can have a certain importance, but are not the major source.
Response: This sentence has been removed and replaced with lines 64-67 in the revised manuscript
Observation: Thanks
In the M&M: Why did you collected 23 samples in 1 year? I wonder how relevant is to use 1 isolate from cattle; 1 hands; 2 environment... Can you give more information about the reasoning behind the sample strategy?
Response: 23 NTS isolates did not represent the sample size. The sample size was n=830 wet swabs from which 23 NTS were isolated (see lines 359-363 of the revised manuscript)
Observation: Thanks
In the M&M: I am aware of the fact that most of the softwares for bioinformatics are "black boxes", but can you explain a bit each analysis you did? For instance, does PathogenFinder 1.1 use an artificial intelligence algorithm? The same for other algorithms used here.
Response: The algorithms for each of the softwares used for bioinformatics have been provided (see lines 398-404 of the revised manuscript)
Observation: Thanks
In the results: Lines 79-85: Be consistent with the number of decimals. Also be consistent with the use of % symbol
Response: Consistency has been observed (lines 98-104, 197, 199, 201, 341 in the revised manuscript)
Observation: Thanks
In the results: Line 95: Not sure if you really meant de-tected or detected.
Response: I meant detected (see lines 135 of the revised manuscript)
Observation: Thanks
In the results: Line 132: Not sure if you really meant pre-dicted or predicted
Response: I meant predicted (see line 188)
Observation: Thanks
In the discussion: Lines 169-171: Not the best way to put that. your findings highlight the need for a deep knowledge on the relevance of food chain on Salmonella sp. AMR. Further studies include in-depth surveys, longitudinal studies and risk assessment models
Response: This section has been rephrased (see lines 220-223 of the revised manuscript)
Observation: Thanks
In the discussion: Lines 194- 200: Be consistent with the genes names, sometimes they are italicized, sometimes not.
Response: The names of genes have been italicized (see lines 132, 139, 142, 261-262, 281, 284, 353 of the revised manuscript)
Observation: Thanks
In the discussion: Lines 228: there is an extra space (enter) there
Response: The extra space has been removed (see line 289 of the revised manuscript)
Observation: Thanks
In the discussion: Lines 239: 50% of 2 is 1, correct? You should probably mention it.
Response: This has been corrected (see line 299 of the revised manuscript)
Observation: Thanks
In the discussion: Lines 241-234: Not sure the meaning of this sentence.
Response: This sentence has been rephrased (see lines 301-303 of the revised manuscript)
Observation: Thanks
In the conclusion: Lines 346-349: I do not agree with this sentence. This study calls for more information about the potential of the food chain in AMR spread in Cameroon.
Response: The sentence has been rephrased (see lines 423-429 of the revised manuscript)
Observation: Thanks
Round 2
Reviewer 1 Report
The authors added the statistical analysis section in the materials and methods section, however, I do not see any statistical analysis within their results, Also, all the tables are not within the frame of the pages, please correct them. The conclusion needs to move after the discussion
Author Response
Reviewer #1:
The authors added the statistical analysis section in the materials and methods section, however, I do not see any statistical analysis within their results, Also, all the tables are not within the frame of the pages, please correct them. The conclusion needs to move after the discussion
Response:
Statistical analysis is now within the results (See lines 120-121, Table 3, lines 186-187, Table 6, line 194).
Tables 2 and 4, initially outside the frame of the pages have been rearranged in landscape format (See Tables 2 and 4) for a clear presentation of the results.
The conclusion has been moved just after the discussion (See lines 363-383).
The conclusion has been improved (See lines 364-365, 373-376)
Reviewer 2 Report
The authors addressed all issues from the first version.
Author Response
Reviewer #2:
The authors addressed all issues from the first version.
Response:
However, the research design has been improved (See 387-390)